# Nonwearable Sensor-Based In-Home Assessment of Subtle Daily Behavioral Changes as a Candidate Biomarker for Mild Cognitive Impairment

**DOI:** 10.3390/jpm12010011

**Published:** 2021-12-24

**Authors:** Takao Yamasaki, Shuzo Kumagai

**Affiliations:** 1Kumagai Institute of Health Policy, Fukuoka 816-0812, Japan; kumagai.shuzo.296@m.kyushu-u.ac.jp; 2Department of Neurology, Minkodo Minohara Hospital, Fukuoka 811-2402, Japan; 3School of Health Sciences at Fukuoka, International University of Health and Welfare, Fukuoka 831-8501, Japan

**Keywords:** mild cognitive impairment, Alzheimer’s disease, nonwearable sensor-based in-home assessment, daily behavior, digital technologies

## Abstract

Patients show subtle changes in daily behavioral patterns, revealed by traditional assessments (e.g., performance- or questionnaire-based assessments) even in the early stage of Alzheimer’s disease (AD; i.e., the mild cognitive impairment (MCI) stage). An increase in studies on the assessment of daily behavioral changes in patients with MCI and AD using digital technologies (e.g., wearable and nonwearable sensor-based assessment) has been noted in recent years. In addition, more objective, quantitative, and realistic evidence of altered daily behavioral patterns in patients with MCI and AD has been provided by digital technologies rather than traditional assessments. Therefore, this study hypothesized that the assessment of daily behavioral changes with digital technologies can replace or assist traditional assessment methods for early MCI and AD detection. In this review, we focused on research using nonwearable sensor-based in-home assessment. Previous studies on the assessment of behavioral changes in MCI and AD using traditional performance- or questionnaire-based assessments are first described. Next, an overview of previous studies on the assessment of behavioral changes in MCI and AD using nonwearable sensor-based in-home assessment is provided. Finally, the usefulness and problems of nonwearable sensor-based in-home assessment for early MCI and AD detection are discussed. In conclusion, this review stresses that subtle changes in daily behavioral patterns detected by nonwearable sensor-based in-home assessment can be early MCI and AD biomarkers.

## 1. Introduction

About 55 million people worldwide suffer from dementia. The proportion of elderly people in the population is increasing in almost every country, and this number is expected to increase to 78 and 139 million in 2030 and 2050, respectively. Dementia has various effects (e.g., physical, psychological, social, and economic) not just on the person with dementia but also on caregivers, families, and society as a whole [1]. Therefore, how to deal with dementia is one of the most important issues worldwide. Moreover, Alzheimer’s disease (AD) is the most common cause of dementia, accounting for an estimated 60–80% of cases [2]. Mild cognitive impairment (MCI) is known to be the prodromal AD stage and is characterized by the loss of cognitive function in one or more cognitive domains, without fulfilling the diagnostic dementia criteria [3]. Short-term memory impairment, disorientation, and visuospatial deficits are the main symptoms in patients with MCI and very mild AD [4,5]. Interestingly, patients with MCI and very mild AD have subtle changes in their daily behavioral patterns alongside these main symptoms [6,7,8]. Thus, subtle changes in daily behavioral patterns may be an indicator of early MCI and AD detection.

Traditional methods (i.e., performance-based assessment and informant-based and self-assessment questionnaires) have been commonly used to assess changes in daily behavioral patterns in MCI and AD for many years [6]. For example, a study with self-assessment questionnaires demonstrated decline in the ability to use the telephone, prepare meals, take medication, manage belongings, keep appointments, talk about recent events, and perform leisure activities/hobbies in patients with MCI [9]. Moreover, the progress of digital technologies (i.e., wearable and nonwearable sensor-based assessment) in various fields has been remarkable recently. Attempts to detect changes in daily behavioral patterns of patients with MCI and AD using digital technologies have also been made in the field of dementia research [10,11,12,13,14]. For example, several studies exhibited a decrease in walking speed, overall activity, and outdoor time in patients with MCI and AD using nonwearable sensors (passive infrared motion sensors and magnetic contact door sensors, and so on) attached to the subject’s home [10,11]. The greatest advantage of digital technologies is that information on subtle changes in behavioral patterns in the patient’s real life at home can be continuously, objectively, and quantitatively collected and evaluated compared with traditional methods [10]. Therefore, detecting subtle behavioral changes in patients with MCI and AD using digital technologies (i.e., wearable and nonwearable sensor-based assessment) can be a new early diagnostic biomarker for MCI and AD.

Generally, two main sensor categories (e.g., wearable and nonwearable sensors) are used to monitor human behavior [10,11,12,13,14]. While wearable sensors allow higher localization accuracy and can detect body movements and vital health metrics, nonwearable sensors are considered less intrusive and do not require any interaction on the user’s side [13]. For the elderly (especially patients with MCI and AD), nonwearable sensors may be less burdensome and more acceptable. Accordingly, this review focused on studies that assessed changes in daily behavioral patterns in patients with MCI and AD using nonwearable sensor-based in-home assessment. This review first describes previous studies on the assessment of behavioral changes in MCI and AD using traditional performance- or questionnaire-based assessments. An overview of previous studies on the assessment of behavioral changes in MCI and AD using nonwearable sensor-based in-home assessment is then provided. Finally, the usefulness and problems of nonwearable sensor-based in-home assessment for early MCI and AD detection are discussed. Thus, this review stresses that subtle changes in daily behavioral patterns detected by nonwearable sensor-based in-home assessment can be early MCI and AD biomarkers.

## 2. Changes in Daily Behavioral Patterns in MCI and AD Using Performance- and Questionnaire-Based Assessments

### 2.1. Definition of Daily Bahavior and Traditional Assessment Methods to Evaluate Daily Bahavioral Changes

The term “daily behavior” in this review includes both basic and instrumental activities of daily living (ADL). Basic ADL are skills required to manage one’s basic physical needs and includes the following categories: ambulating, feeding, dressing, personal hygiene, continence, and toileting. In contrast, instrumental ADL requires more complex thinking skills, including organization skills. This includes transportation and shopping, managing finances, shopping and meal preparation, house cleaning and home maintenance, managing communication with others, and managing medications [15].

The Barthel Index and the Katz Index of ADL are the best-known tools for basic ADL assessment in dementia [16]. The Barthel Index is the most commonly used instrument and is a 10-item outcome measure that is completed by a healthcare professional. The Katz Index is a six-item measurement that also has to be completed by a healthcare professional. Moreover, many instruments for evaluation of instrumental ADL exist. The three established methods to assess instrumental ADL are as follows: performance-based assessment, informant-based questionnaire, and self-assessment questionnaire [17]. Each assessment method has its strengths and weaknesses (Table 1). The performance-based assessment provides an objective behavioral evaluation of functional skills by a trained rater. Nevertheless, it is a time consuming and costly assessment, and only a restricted number of activities can be evaluated. Another limitation is the difference in patients’ performance between artificial (clinical) settings and their performance at home [17]. An informant or proxy can be a spouse, partner, relative, or close friend for the informant-based questionnaire. Possible disadvantages are the informants’ characteristics (e.g., anxiety, depression, caregiver burden, and general perceived health) that may influence informant ratings. In contrast, advantages include the ease of administration, ratings based on real-world functional instrumental ADL performance, and the fact that the patient is not burdened by an assessment. These advantages make the informant-based questionnaire the most commonly used instrumental ADL assessment method in dementia evaluation [17]. A self-assessment questionnaire is the easiest method. However, impaired insight can make the reports invalid in patients with dementia [17].

### 2.2. Changes in Daily Bahavioral Patterns Observed in MCI and AD Based on Performance- and Questionnarie-Based Assessments

Many reports evaluating daily behavioral changes in individuals with MCI and AD using performance-based assessments or informant-based or self-assessment questionnaires have been noted [6]. Regarding the performance-based instruments, the Direct Assessment of Functional Status was the best measure for detecting differences in global instrumental ADL functioning between MCI and healthy controls [6]. This measure is a standardized observation-based checklist designed to assess the functional capabilities of adults with AD, dementia, and schizophrenia. The examiner needs the evaluation form, pen or pencil, and ADL materials for testing. Simulated daily tasks are observed in the seven following areas: time orientation, communication, transportation, finance, shopping, grooming, and eating [18]. Using the Direct Assessment of Functional Status, Pereira et al. [19] found that patients with MCI performed significantly worse than healthy controls and better than patients with AD. Financial and shopping skills were the items that differentiated patients with MCI from healthy controls.

For the informant-based questionnaire, the Alzheimer’s Disease Cooperative Study scale for ADL in MCI seems to be a useful tool for global instrumental ADL assessment [6]. This questionnaire assesses the competence of patients with MCI in basic and instrumental ADL (covering 18 areas). It can be completed by a caregiver in a questionnaire format or administered by a clinician/researcher as a structured interview with caregivers [20]. Moreover, Perneczky et al. [21] used this questionnaire for measuring instrumental ADL in MCI. The overall score of this scale was significantly lower in the MCI group where the impaired ADL (14 out of 18 activities) were found. Activities involving memory or complex reasoning were particularly impaired, whereas more basic activities were unimpaired. In another paper, Perneczky et al. [22] examined whether this scale could be a significant predictor of the MCI diagnosis. They demonstrated that this scale discriminated well between patients and healthy controls with a sensitivity and specificity of 0.89 and 0.97, respectively, using receiver operator curve analysis.

The Seoul-Instrumental ADL and Lawton and Brody’s Instrumental ADL were used for the self-assessment questionnaires [23,24]. For example, Pérès et al. [24] assessed for instrumental ADL (telephone, transport, medication, and finances) in patients with MCI and dementia using Lawton and Brody’s Instrumental ADL. Patients with MCI were more frequently instrumental ADL-restricted (34.3%) than healthy controls (5.4%) but less than those with dementia (91.1%). Interestingly, the instrumental ADL-restricted subjects with MCI were more likely to develop dementia in >2 years (30.7%) than those with non-instrumental ADL-restricted MCI (7.8%) [24]. In addition, the odds ratios for dementia were 7.4 and 2.8 in instrumental ADL-restricted and non-instrumental ADL-restricted MCI, respectively, compared with healthy controls [24].

The instrumental ADL deficits were also analyzed between MCI subtypes. Moreover, MCI can be classified according to the presence/absence of episodic memory impairments (amnestic or non-amnestic) and the number of affected cognitive domains (single or multiple domains) [3]. A systematic review exhibited that the instrumental ADL deficits tended to be more pronounced in amnestic MCI than in non-amnestic MCI. The instrumental ADL deficits were more pronounced in the multiple-domain MCI than in the single-domain MCI [6].

Overall, changes in daily behavior are likely to be consistently present even in individuals with MCI in both the performance- and questionnaire-based methods. Furthermore, patients with MCI with instrumental ADL deficits seem to have a higher risk of converting to dementia than patients without ADL deficits. Thus, assessment of daily behavior (in particular, the instrumental ADL) is useful for early MCI detection and prognosis prediction.

Concerning the nature of changes in daily behavioral patterns (instrumental ADL deficits), Bruderer-Hofstetter et al. [8] recently developed a comprehensive model of ADL functioning that depicts the relevant influencing factors. In their studies, various factors are thought to be involved in these functional changes in patients with MCI. The relevant influencing factors include five cognitive factors (i.e., memory, attention, executive function, and two executive function subdomains (problem solving/reasoning and organization/planning)), five physical factors (i.e., seeing functions, hearing functions, balance, gait/mobility functions, and functional mobility functions), two environmental factors (i.e., social network/environment and support of social network/environment), and one personal factor (i.e., education) [8].

## 3. Changes in Daily Behavioral Patterns in MCI and AD Based on Nonwearable Sensor-Based In-Home Assessment

### 3.1. Digital Technologies for Monitoring of Daily Behavioral Patterns

In general, two main categories of sensors (i.e., wearable and nonwearable sensors) are used to monitor human behavior [13,14,25]. The two kinds of sensors have been used extensively in various systems. Wearable sensors are usually attached to a person directly (e.g., bracelet or cardio sensors) or to their clothes (e.g., an accelerometer or a step counter) to measure location, pulse rate, body temperature, blood pressure, and other important metrics as well as motion characteristics [13]. Conversely, nonwearable sensors are usually deployed in stationary locations of a house or a room and can detect a person and his movements and activities. Nonwearable sensors can specify the operational status of objects, measure water flow, room temperature, or door/cupboard opening/closings [13]. The types and characteristics of nonwearable sensors are summarized in Table 2.

Wearable sensors have the advantage of higher localization accuracy and tracking. Additionally, wearable sensors are much more available and provide a timely and economical fashion for detecting MCI and AD compared with nonwearable sensors [13]. However, wearable sensor-based monitoring is more intrusive and demands that older adults with various degrees of cognitive levels to remember wear the devices as well as the need for regular charging of the devices [11]. In contrast, nonwearable sensors are less intrusive and can monitor activities in a real life and naturalistic environment without causing any interference to an individual’s daily routines [11]. From these characteristics, it seems that nonwearable sensors are more suitable for the monitoring and detection of patients with MCI and AD. Thus, we focused on research using nonwearable sensor-based in-home assessment.

### 3.2. Changes in Daily Bahavioral Patterns Observed in MCI and AD Based on Nonwearable Sensor-Based In-Home Assessment

Several studies have investigated one of the daily behaviors (e.g., walking, sleeping, and going out) in patients with MCI and AD (Table 3). For example, Hayes et al. [26] investigated the walking speed and daily activity of patients with MCI and healthy elderly people living independently and alone in the community using unobtrusive sensors (passive infrared motion sensors for each room and magnetic contact door sensors for each door) at home for at least 6 months. Walking speed was more variable in the MCI group than in the healthy group. In addition, the day-to-day activity pattern was more variable in patients with MCI. Dodge et al. [27] also examined in-home walking speeds using passive infrared sensors fixed in a series on the ceiling of homes of patients with amnestic MCI and non-amnestic MCI and healthy controls for >3 years. Patients with non-amnestic MCI were characterized by a reduced walking speed. Furthermore, two distinct trajectories (the highest and lowest variability) were predominantly associated with non-amnestic MCI. These studies suggest the alteration of walking behavior in patients with MCI. Moreover, Hayes et al. [28] explored the relationship between sleep disturbances and patients with MCI in community-dwelling seniors using wireless passive infrared motion sensors in each room of the home (bedroom, bathroom, kitchen, living room, and hallway-entry areas) and magnetic contact door sensors for each door over 6 months. Consequently, patients with amnestic MCI and non-amnestic MCI, and cognitively intact volunteers showed different patterns of sleep disturbance-. In particular, patients with amnestic MCI had less disturbed sleep than both those with non-amnestic MCI and healthy subjects. These differences in sleep disruption between amnestic MCI and non-amnestic MCI may be related to differences in the pathology underlying these MCI subtypes. Petersen et al. [29] investigated the relationship between time out-of-home and cognitive status, physical ability, and emotional state in patients with MCI and healthy elderly using pyroelectric infrared motion sensors in each room and contact sensors on the refrigerator and doors of the home for up to 1 year. They found that cognitive status was significantly associated with time out-of-home. Furthermore, patients with MCI spent an average of 1.67 h more inside the home than healthy elderly.

Recent studies focused on changes in daily behavioral patterns in general, rather than just one daily activity (Table 3). Urwyler et al. [30] investigated differences in daily behavioral pattern performance between patients with dementia and healthy controls using unobtrusive sensors for 20 consecutive days. An unobtrusive sensor network comprising 10 wireless sensor boxes was installed in the home (Figure 1). Each sensor box consisted of five sensors (temperature, humidity, luminescence, presence (passive infrared radiation), and acceleration). Consequently, a significant difference in daily behavioral patterns was observed between patients with dementia and healthy controls. Specifically, patients with dementia revealed unorganized behavior patterns (Figure 2). Rawtaer et al. [31] examined changes in behaviors in patients with MCI using the multiple sensor system (passive infrared motion sensors, proximity beacon tags, a sensor-equipped medication box, a bed sensor, and a wearable sensor) for >2 months. Patients with MCI were less active than healthy subjects and had more sleep interruptions per night. In addition, patients with MCI had forgotten their medications more times per month than healthy subjects. Overall, changes in various kinds of daily behavioral functions were observed even in patients with MCI.

### 3.3. Machine Learning–Based Prediction Model for Detecting Individuals with MCI

Machine learning is a subdiscipline of artificial intelligence and has been extensively used in recent studies to predict behavioral/cognitive abnormalities utilizing sensor-based activity data [32,33]. Several studies have been noted to apply these methods to MCI differentiation [34,35,36,37] (Table 3). Moreover, support vector machine and random forest were the most commonly used techniques, although a wide variety of machine learning techniques were employed. Several metrics are used to evaluate prediction models (e.g., area under the receiver operator characteristic and precision–recall curves) and the F-score [32,33,34,35,36,37].

**Table 3 jpm-12-00011-t003:** Previous studies assessing daily behavioral patterns in patients with MCI and AD using nonwearable sensor-based in-home assessment.

References	Participants and Study Protocol (1. Study Design; 2. Participants; 3. Sensor Type; 4. Duration; 5. Machine Learning Technique)	Main Findings
Hayes et al. [26]	Observational cross-sectional study Healthy (*n* = 7; mean age: 90.0 y; F:M = 5:2); MCI (*n* = 7; mean age: 88.4 y; F:M = 4:3) Passive infrared motion sensors and magnetic contact door sensors Six months	- Walking speed was more variable in patients with MCI. - Day-to-day pattern of activities was more variable in patients with MCI.
Dodge et al. [27]	Observational longitudinal study Healthy (*n* = 54; mean age: 84.9 y; %F: 91%); amnestic MCI (*n* = 8; mean age: 84.5 y; %F: 88%); non-amnestic MCI (*n* = 31; mean age: 83.8 y; %F: 84%) Passive infrared sensors Three years	- Daily walking speeds and their variability were associated with non-amnestic MCI.
Hayes et al. [28]	Observational cross-sectional study Healthy (*n* = 29; mean age: 87.5 y; F:M = 26:3); amnestic MCI (*n* = 6; mean age: 84.8 y; F:M = 5:1); non-amnestic MCI (*n* = 10; mean age: 86.5 y; F:M = 9:1) Wireless passive infrared motion sensors and magnetic contact door sensors Six months	- Patients with amnestic MCI showed less sleep disturbance than both those with non-amnestic MCI and healthy elderly.
Petersen et al. [29]	Observational study Healthy (*n* = 75; mean age: not clear; F:M = not clear); MCI (*n* = 10; mean age: not clear; F:M = not clear) Pyroelectric infrared motion sensors and contact sensors One year	- Patients with MCI spent an average 1.67 h more inside the home than healthy elderly.
Urwyler et al. [30]	Observational study Healthy (*n* = 10; mean age: 73.9 y; F:M = 7:3); dementia (*n* = 10; mean age: 76.7 y; F:M = 7:3) A wireless-unobtrusive sensors (temperature, humidity, luminescence, presence [passive infrared radiation], and acceleration) Twenty consecutive days	- Patients with dementia showed unorganized behavior patterns.
Rawtaer et al. [31]	Observational cross-sectional study Healthy (*n* = 21; mean age: 73.0 y; F:M = 14:7); MCI (*n* = 28; mean age: 75.1 y; F:M = 19:9) Multiple sensor system (passive infrared motion sensors, proximity beacon tags, a sensor equipped medication box, a bed sensor, and a wearable sensor) Two months	- Patients with MCI were less active than healthy subjects and had more sleep interruptions per night. - Patients with MCI had forgotten their medications more times per month than healthy subjects.
Akl et al. [34]	Observational longitudinal study Healthy (*n* = 79; mean age: not clear; F:M = not clear); MCI (*n* = 18; mean age: not clear; F:M = not clear) Passive infrared motion sensors and wireless contact switches Three years Support vector machine, random forest	- Variabilities in weekly walking speed, morning and evening walking speeds, and subjects’ age and gender were the most important for the process of detecting MCI. - This study autonomously detected MCI with receiver operating characteristic curve (0.97) and precision–recall curve (0.93) using a time windows of 24 weeks.
Akl et al. [35]	Observational longitudinal study Healthy (*n* = 59; mean age: not clear; F:M = not clear); amnestic MCI (*n* = 11; mean age: not clear; F:M = not clear); non-amnestic MCI (*n* = 15; mean age: not clear; F:M = not clear) Passive infrared motion sensors and wireless contact switches Three years Clustering (affinity propagation)	- This study automatically detected MCI (F0.5 score, 0.856) and non-amnestic MCI (F0.5 score, 0.958).
Alberdi et al. [36]	Observational longitudinal study Healthy (*n* = 13; mean age: 82.85 y; F:M = 9:4); at risk (*n* = 10; mean age: 86.20 y; F:M = 10:3); MCI (*n* = 6; mean age: 84.50 y; F:M = 5:1) Passive infrared motion sensors Two years Regression: support vector regression, linear regression, *K* nearest neighbors; Classification: support vector machine, adaboost, multilayer perceptron, random forest	- Sleep and overnight patterns along with daily routine features contributed to the prediction of several health assessments. - All algorithms could build statistically significant prediction models.
Nakaoku et al. [37]	Observational study Normal cognition (*n* = 55; mean age: 75.0 y; F:M = 18:37); cognitive impairment (*n* = 23; mean age: 78.0 y; F:M = 6:17) Unobtrusive in-house power monitoring system (air conditioner, microwave oven, washing machine, rice cooker, television, and induction heater) One year Generalized linear model	- Three independent power monitoring parameters (air conditioner, microwave oven, and induction heater) representing activity behavior were associated with cognitive impairment. - The prediction model with power monitoring data had better predictive ability (accuracy, 0.82; sensitivity, 0.48; and specificity, 0.96).

ADL activities of daily living, MCI mild cognitive impairment, AD Alzheimer’s disease.

Akl et al. [34] explored the ability of signal processing along with machine learning algorithms to autonomously detect MCI using home-based unobtrusive sensing technologies (passive infrared motion sensors in rooms, wireless contact switches on doors, and motion sensors on the ceiling). They found that variabilities (i.e., weekly walking speed, morning and evening walking speeds, and subjects’ age and gender) were the most important factors in the process of detecting MCI. The authors autonomously detected MCI with receiver operator characteristic and precision–recall curves of 0.97 and 0.93, respectively, using a time window of only 24 weeks [34]. A clustering-based method to automatically detect MCI using estimated generalized linear models of their home activity was proposed by Akl et al. in another study [35]. Continuous monitoring was conducted via unobtrusive sensing technologies (passive infrared motion sensors in the room and wireless contact switches on the doors). The authors automatically detected MCI and non-amnestic MCI with F0.5 scores of 0.856 and 0.958, respectively [35]. Moreover, Alberdi et al. [36] assessed the possibility of detecting changes in psychological, cognitive, and behavioral symptoms of MCI by making use of unobtrusively collected smart home behavior data and machine learning techniques. They found that sensor-based activity observations (e.g., sleep and overnight patterns) and daily routine features contributed significantly to the prediction of several health assessments. All algorithms could build statistically significant prediction models [36]. In addition, Nakaoku et al. [37] investigated whether unobtrusive in-house power monitoring technologies could be used to predict cognitive impairment. Daily activity data were collected using a well-established unobtrusive in-house power monitoring system installed in the participants’ homes. Several electric appliances (air conditioner, microwave oven, washing machine, rice cooker, television, and induction heater) were monitored. Three independent power monitoring parameters (air conditioner, microwave oven, and induction heater) representing activity behavior were associated with cognitive impairment. Regarding the prediction models for cognitive impairment, the model with power monitoring data had a better predictive ability (accuracy, 0.82; sensitivity, 0.48; and specificity, 0.96) than the model without power monitoring data (accuracy, 0.76; sensitivity, 0.30; and specificity, 0.95) [37]. From the findings of these studies, combining data collection by sensors and machine learning are useful to detect patients with MCI.

## 4. Discussion

Previous studies that evaluated the alteration of daily behavioral patterns in patients with MCI and AD using traditional performance- and questionnaire-based [6,7,8,9,21,22,23,24] (Section 2.2) and nonwearable sensor-based in-home assessments [10,11,26,27,28,29,30,31,34,35,36,37] (Section 3.2 and Section 3.3; Table 3) were described in this review. From the findings of these studies, all these assessment methods are considered useful in differentiating between patients with MCI and healthy elderly people. However, traditional assessment has various weaknesses (e.g., being influenced by subjective judgment and deriving results dissociated from the situation of daily behavior in real life (home life) [17]; Section 2.1; Table 1). On the other hand, the evaluation of daily behavior using nonwearable sensor-based in-home assessment has various advantages over traditional assessment (Table 1). First, the status of daily behavior can be observed continuously for a long period. Second, the status of daily behavior can be evaluated in the natural home environment. Third, it is more objective than a questionnaire that contains the subjectivity of the person or caregiver [10,11]. Furthermore, a machine learning–based predictive model has succeeded in identifying MCI from healthy elderly by combining data obtained from sensors [11,34,35,36,37]. Thus, nonwearable sensor-based in-home assessment for daily behavior is believed to be an objective, quantitative, and realistic method that replaces or assists traditional assessment methods.

As mentioned earlier, the prediction model to discriminate between patients with MCI and healthy elderly based on changes in daily behavioral patterns on nonwearable sensors seems to be good. However, some problems which need to be solved before practical use were noted. First, reducing the number of sensors to reduce labor and cost is necessary [11,31,37]. Second, developing sensors that are easy to use in any home type is needed [11,37]. Third, using sensors that allow data to be standardized in any living environment (e.g., monitoring with a single sensor in a single location) is needed. Fourth, using big data and artificial intelligence is desirable to improve the ability to distinguish between patients with MCI and healthy elderly [10,11,31,37]. Fifth, accumulating not only cross-sectional data but also longitudinal data is desirable [11,31]. If these points can be solved, nonwearable sensor-based in-home assessment for daily behavioral changes may become the best digital MCI and AD biomarker. To solve these problems, the laboratory of the current study supervised a study on a technique that can identify changes in toilet behavior of elderly subjects using an artificial intelligence light sensor (supervised by S.K.) [38]. Moreover, a previous study [30] reported changes in daily behavioral patterns (i.e., the appearance of unorganized daily behavioral patterns) in patients with dementia. In this study, it seemed that patients with dementia used the toilet more during the day than healthy elderly people (Figure 2). Thus, identifying changes in daily behavioral patterns in patients with MCI with toilet sensors alone without the use of multiple sensors may be possible. The current study succeeded in capturing toilet behavioral changes in the elderly living alone using artificial intelligence light sensors and has put them into practical use. In addition, a system that enables the distinction between patients with MCI and healthy individuals through toilet behavior changes will be developed soon.

Point-of-care testing (defined as a test performed outside a central laboratory [39]) at home is advancing with regard to various diseases with the development of digital technologies in recent years, [40,41,42]. However, the diagnosis of dementia is mainly performed in hospitals or medical laboratories using various diagnostic biomarkers (e.g., neuropsychological, neurophysiological, biological, and genetic biomarkers) [43,44,45,46,47]. Using these biomarkers as point-of-care diagnostic tools is difficult because biomarkers are invasive, expensive, and can only be tested in medical institutions. No studies exist on applying nonwearable sensors to point-of-care testing in patients with dementia. However, nonwearable sensor-based (including our toilet sensor) in-home assessment is applied to point-of-care testing for early detection of MCI and AD. Furthermore, the development of nonwearable sensor-based in-home assessment can provide the best intervention method for improving individual quality of life. Indeed, nonwearable sensor-based in-home assessment is useful for personalized medicine in patients with MCI and AD.

## 5. Conclusions

Subtle changes in daily behavioral patterns are observed along with impairments of short-term memory, orientation, and visuospatial perception from the early stage of the disease in patients with MCI and early AD. Thus, subtle changes in daily behavioral patterns can be important indicators of early MCI and AD detection.

Both traditional performance- or questionnaire-based assessment and sensor-based in-home assessment are useful to evaluate the daily behavioral patterns in patients with MCI and AD, whereas sensor-based in-home assessment is more objective and quantitative. Furthermore, subtle changes in daily behavioral patterns at home rather than in an extraordinary environment (e.g., hospitals or medical laboratories) can be captured using various nonwearable sensors that do not make the person aware that he/she is being monitored. In conclusion, nonwearable sensor-based in-home assessment can be an early diagnostic biomarker for MCI and AD.

Developing a system that can easily capture subtle changes in people’s daily behavioral patterns at home by using a single sensor instead of multiple sensors will be necessary in the future. Such a system could be a point-of-care testing system providing an excellent early diagnosis of MCI and AD.

## Figures and Tables

**Figure 1 jpm-12-00011-f001:**
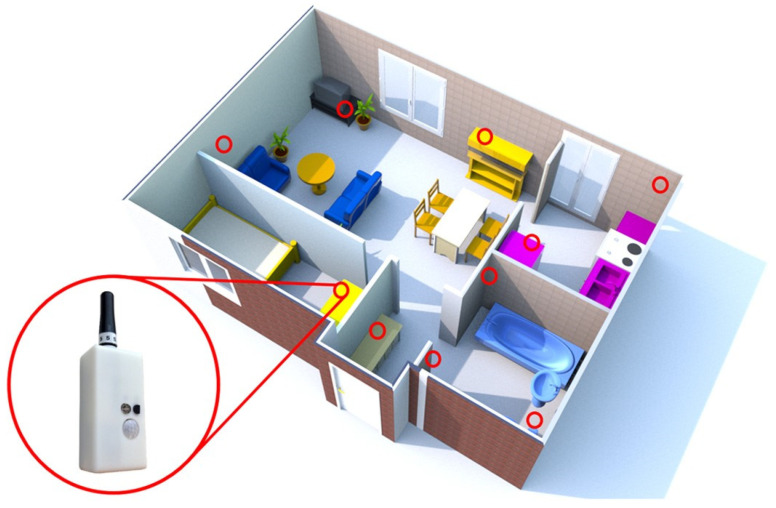
Floor plan of an apartment showing the placement of sensor boxes (red circles). The figure is adapted from Urwyler et al. [30] (CC BY 4.0).

**Figure 2 jpm-12-00011-f002:**
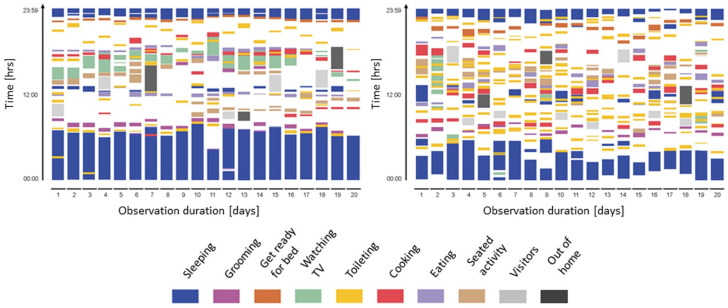
Activity maps of a healthy control (**left**) and a patient with dementia (**right**) visualized from data continuously measured for 20 consecutive days. Activity maps of patients with dementia reveal unorganized behavior patterns, and heterogeneity differed significantly between the healthy control and the patient. The figure is adapted from Urwyler et al. [30] (CC BY 4.0).

**Table 1 jpm-12-00011-t001:** Assessment methods for daily behavioral patterns in patients with MCI and AD.

Assessment Methods	Characteristics	Strengths	Weaknesses
Performance-based assessment	- Behavioral evaluation by a trained rater	- More objective than the questionnaire method	- Time consuming - Expensive - Only a limited number of activities can be evaluated - It does not always reflect the actual ADL at home
Informant-based questionnaire	- Questionnaire method completed by a suitable informant	- Easier than performance-based assessment - More objective than a self-assessment questionnaire	- The results are influenced by the person’s physical and mental conditions
Self-assessment questionnaire	- Questionnaire method completed by the patient himself/herself	- The easiest method	- Results are not always accurate because of cognitive decline
Nonwearable sensor-based in-home assessment	- Behavioral evaluation by various sensors installed at home	- More objective and quantitative than other methods	- Expensive - It takes time and effort to install sensors, monitor behavior, and analyze results

ADL activities of daily living, MCI mild cognitive impairment, AD Alzheimer’s disease.

**Table 2 jpm-12-00011-t002:** Types and characteristics of nonwearable sensors.

Sensor	Measurement Type	Characteristics
Infrared sensors	Motion	- Most frequently used nonwearable sensors - Discover human presence in a room - Detect motion in a specific area - Locate a human within a house
Ultrasonic sensors	Motion	- Person detection and localization by measuring distances to objects
Photoelectric sensors	Motion	- Detect a light source and output a signal
Vibration sensors	Vibration	- Detect a person falling, interaction with various objects, flushing toilets, and water flows
Pressure sensors	Pressure on object	- Detect the presence of a person, steps, and fall events - Deploy in the form of floor mats and smart tiles
Magnetic switches	Opening or closing	- Detect opening and closing of doors or cupboards - Provide information on users accessing particular rooms and opening dressers, refrigerators, or trash cans
Audio sensors	Activity-related sound	- Detect sounds in a house - Discriminate between different types of sounds
Wattmeter and other sensors	Consumption information	- Measure electricity consumption of domestic appliances and light

## Data Availability

Not applicable.

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
