# Peer review of "Nonwearable Sensor-Based In-Home Assessment of Subtle Daily Behavioral Changes as a Candidate Biomarker for Mild Cognitive Impairment"

_jpm, 2021, doi:10.3390/jpm12010011_

Round 1
Reviewer 1 Report
Reviewer comments:
Title: Digital biomarkers for mild cognitive impairment: evidence from changes in activities of daily living
ID: jpm-1486139
(a) ARTICLE RANKING
* Good
(b) RECOMMENDATION
* Minor revision.
(c) Comments of Reviewer
TITLE:
* what do you mean by Digital biomarkers; how about the term digital tools?
ABSTRACT:
* A short form for activities of daily living (ADL) is uncommon. First, define the concept.
INTRODUCTION
* It seems too short, bring more connectivity and discuss recent developments in literature.
MAIN TEXT
* Consider relevant citations as many as possible.
* Elaborate text of subsection 3.1. Digital technologies for ADL monitoring
Consider Point-of-Care subsection
* Elaborate section 4. Discussion
CONCLUSIONS
* It seems short and consider the advantages of ADL, digital tools, and future implications.
Author Response
To Reviewer 1:
Comment 1: TITLE: * what do you mean by Digital biomarkers; how about the term digital tools?
Response: Following your suggestion and the comment 5 of Reviewer 4, we have changed the title to a more appropriate one.
Comment 2: ABSTRACT: * A short form for activities of daily living (ADL) is uncommon. First, define the concept.
Response: Since the term of “activities of daily living (ADL)” seems inappropriate, we changed the term of “activities of daily living (ADL)” to “daily behavior”.
Comment 3: INTRODUCTION * It seems too short, bring more connectivity and discuss recent developments in literature.
Response: Thank you for your valuable suggestion. We have rewritten the Introduction section extensively by citing the appropriate paper (P 1, Para 1–P 2, Para 3).
Comment 4: MAIN TEXT * Consider relevant citations as many as possible. * Elaborate text of subsection 3.1. Digital technologies for ADL monitoring. Consider Point-of-Care subsection. * Elaborate section 4. Discussion.
Response: Following your suggestion, we have elaborated the main text by adding appropriate citations. We have added things about point-of-care testing to the Discussion section (P 11, Para 2).
Comment 5: CONCLUSIONS * It seems short and consider the advantages of ADL, digital tools, and future implications.
Response: Following your suggestion, we have described these matters in the Conclusion section (P 11, Para 3–5).
Reviewer 2 Report
The submitted manuscript is written in an understandable and well-accepted form.
Author Response
To Reviewer 2:
Comment 1: The submitted manuscript is written in an understandable and well-accepted form.
Response: Thank you for your peer review.
Reviewer 3 Report
Digital biomarkers show definite promise in detecting MCI. Timely detection of patients with MCI is necessary to provide support and devise a (non) pharmaceutical management approach. In this review, Takao et al compared traditional performance- or questionnaire-based assessments with house sensor-based assessments and discussed the recent progress and challenging. This review is well-written and well-organized. Few additional content are needed to enhance the significance to a larger community.
- The reason that the authors omit the content of wearable sensors is not acceptable. Based on the title, a thorough summary of digital technologies for ADL assessment is needed. Compared with nonwearable sensors in smart house, mobile and wearable digital devices are much more available and provide a timely and economical fashion for detecting MCI and AD (Lampros C. Kourtis et al, 2019). The author need to add particular content about the mobile and wearable digital devices for detecting MCI and AD.
- More details about nonwearable sensors are necessary such as category and sensor selection (Christian et al, 2016).
- Table 2: more information of Participants are needed like age and gender.
- Table 2: the informationof Machine learning technique is lacking in some listed studies
- A paragraph discussing future directions is suggested.
Author Response
To Reviewer 3:
Comment 1: The reason that the authors omit the content of wearable sensors is not acceptable. Based on the title, a thorough summary of digital technologies for ADL assessment is needed. Compared with nonwearable sensors in smart house, mobile and wearable digital devices are much more available and provide a timely and economical fashion for detecting MCI and AD (Lampros C. Kourtis et al, 2019). The author need to add particular content about the mobile and wearable digital devices for detecting MCI and AD.
Response: According to your suggestion, we have increased the description of wearable sensors by citing the paper recommended by you (P 2, Para 3; P 5, Para 1; P 6, Para 1). However, the main subject of this paper is research on nonwearable sensors, so that the title of this paper has been changed. It was recommended by Reviewer 4 to change the title.
Comment 2: More details about nonwearable sensors are necessary such as category and sensor selection (Christian et al, 2016).
Response: Thank you for your valuable suggestion. We have added more information of nonwearable sensors (P 5, Para 1; Table 2).
Comment 3: Table 2: more information of Participants are needed like age and gender.
Response: We have added more information of participants (age and gender) in Table 3.
Comment 4: Table 2: the information of Machine learning technique is lacking in some listed studies.
Response: We have described the information of Machine learning technique more carefully in Table 3.
Comment 5: A paragraph discussing future directions is suggested.
Response: This comment is related to comments of Reviewer 1 (Comment 5) and Reviewer 4 (Comment 3). According to suggestion from you and reviewers 1 and 4, we have added a paragraph of future direction (P 11, Para 5).
Reviewer 4 Report
Manuscript ID: jpm-1486139
Type of manuscript: Review
Title: Digital biomarkers for mild cognitive impairment: evidence from changes in activities of daily living
Authors: Takao Yamasaki and Shuzo Kumagai
This review describes the studies on activities of daily living assessments of patients with early stage of Alzheimer’s disease using digital technologies.
There are some major problems in the manuscript and a careful review is needed.
- There are several places where sloppy editing is in evidence and English editing is also required. So many abbreviations are confusing, and the flow of reading is disturbed.
- How does this review differ from reference #7?
- Can the authors include the research's future directions in the conclusion section?
- The authors can elaborate more details about digital biomarker in the introduction.
- The authors stated in lines 178–179 “Thus, this review focused on studies that assessed ADL in patients with MCI and AD using nonwearable sensor-based in-home assessment.” If the focus of this review is mostly on non-wearable sensors, this should be indicated in the title.
For all these reasons I do not recommend the publication of this review in the Journal of Personalized Medicine in the present form.
Author Response
To Reviewer 4:
Comment 1: There are several places where sloppy editing is in evidence and English editing is also required. So many abbreviations are confusing, and the flow of reading is disturbed.
Response: English editing of the revised manuscript has been done. We have reduced the abbreviations as much as possible.
Comment 2: How does this review differ from reference #7?
Response: Our paper describes not only research using digital technologies, but also research using traditional methods. Thus, our paper is different from paper of Piau et al. (2019).
Comment 3: Can the authors include the research's future directions in the conclusion section?
Response: This comment is related to comments of Reviewer 1 (Comment 5) and Reviewer 3 (Comment 5). According to suggestion from you and reviewers 1 and 3, we have added a paragraph of future direction (P 11, Para 5).
Comment 4: The authors can elaborate more details about digital biomarker in the introduction.
Response: We have increased the description of digital technologies in the Introduction section (P 2, Para 2–3).
Comment 5: The authors stated in lines 178–179 “Thus, this review focused on studies that assessed ADL in patients with MCI and AD using nonwearable sensor-based in-home assessment.” If the focus of this review is mostly on non-wearable sensors, this should be indicated in the title.
Response: Thank you for your suggestion. Following your suggestion, we have changed the title to a more appropriate one.
Round 2
Reviewer 4 Report
The revised manuscript is an understandable and it will be accepted in the present form.